# *Euscaphis japonica* Kanitz Fruit Exerts Antiobesity Effects by Inhibiting the Early Stage of Adipogenic Differentiation

**DOI:** 10.3390/nu15143078

**Published:** 2023-07-08

**Authors:** Eunbi Lee, Juhye Park, Ju-Ock Nam

**Affiliations:** Department of Food Science and Biotechnology, Kyungpook National University 80, Daehakro, Bukgu, Daegu 702701, Republic of Korea; 21eunbi@knu.ac.kr (E.L.); pdw8606@knu.ac.kr (J.P.)

**Keywords:** *Euscaphis japonica* Kanitz fruit, antiobesity, adipogenesis, 3T3-L1 preadipocytes, mitotic clonal expansion (MCE)

## Abstract

During the worldwide COVID-19 outbreak, there was an increase in the prevalence of obesity, including childhood obesity, due to which the awareness of obesity and interest in treatment increased. Accordingly, we describe EJF (*Euscaphis japonica* Kanitz fruit) extract as a candidate for naturally derived antiobesity agents. In this study, we found that EJF is involved in the early stage of adipogenic differentiation in vitro and finally inhibits adipogenesis. We propose two mechanisms for the antiobesity effect of EJF. First, EJF inhibits MDI-induced mitotic clonal expansion (MCE) by inducing cell cycle arrest at the initiation of adipogenic differentiation. The second aims to regulate stability and activation at the protein level of IRS1, which initiates differentiation in the early stage of differentiation. As a result, it was found that the activation of Akt decreased, leading to the inhibition of the expression of adipogenesis-related transcription factors (PPARγ, C/EBPα) and the subsequent suppression of adipogenic differentiation. In summary, we suggest that EJF can inhibit adipogenesis and lipid accumulation by suppressing the early stage of adipogenic differentiation in 3T3-L1 adipocytes. These findings indicate that EJF’s functionality could be beneficial in the treatment of obesity, particularly childhood obesity associated with adipocyte hyperplasia.

## 1. Introduction

During the worldwide COVID-19 outbreak, there was an increase in the prevalence of obesity due to increased food intake and decreased activity among children, adolescents, and young adults. In particular, the proportion of children and adolescents with obesity aged between 6 and 19 years significantly increased from 4% in 1975 to >18% in 2016, and the number of patients with childhood obesity has recently increased rapidly [1]. As the basic number of adipocytes is formed during childhood and adolescence, childhood obesity is accompanied by hyperplasia of adipocytes [2]. The complications of childhood obesity include hypertension, left ventricular hypertrophy, atherosclerosis, insulin resistance, diabetes, asthma, and nonalcoholic fatty liver, which are similar to those of adult obesity [3]. However, childhood obesity is considered a much greater risk factor than adult obesity. The period of growth and development in adolescence is associated with a normal increase in insulin resistance, but an additional insulin resistance due to obesity during this period causes glucose intolerance and may lead to type 2 diabetes [4]. Furthermore, because the enzymes that synthesize sex hormones are expressed in the adipose tissue, excessive adipose tissue can induce high androgen levels, causing hormonal imbalances. Consequently, girls with childhood obesity may experience menstrual abnormalities such as amenorrhea, irregular bleeding, and polycystic ovary syndrome, which can result in impaired fertility [5].

The currently available drugs for obesity treatment can be broadly classified into two categories, viz. medications that act on the central nervous system to influence appetite and medications that act on either the central nervous system or peripheral systems to promote thermogenesis, as well as drugs that act on the gastrointestinal tract to reduce absorption. However, these drugs can cause side effects, such as insomnia, dry mouth, tachycardia, constipation, and heart valve diseases [6,7]. Hence, there has been an increasing demand for new treatments due to the side effects of commercially available obesity drugs, resulting in the emergence of natural-based antiobesity agents [8,9]. In fact, natural-based antiobesity agents have gained popularity among consumers, particularly due to the general perception that they exert minimal or no side effects compared with pharmaceutical drugs. The commercial market for antiobesity agents derived from natural sources, which have been explored for their efficacy and safety, is of significant magnitude [10]. Therefore, we selected the fruit of *Euscaphis japonica* Kanitz as a candidate for natural-based antiobesity agents. Although the antiobesity effects of the leaf of *E. japonica Kanitz* have been reported, the molecular mechanisms related to adipogenic differentiation have not been completely elucidated [11], and the antiobesity effects of the fruit have not yet been reported. Therefore, in this study, we investigated whether *E. japonica* Kanitz fruit (EJF) extract exerts an inhibitory effect on adipocyte hyperplasia by inhibiting adipogenic differentiation for the treatment of obesity, especially childhood obesity. Through our study, we present EJF as a candidate antiobesity agent derived from natural products.

## 2. Materials and Methods

### 2.1. Preparation of Euscaphis japonica (Thunb.) Kanitz Fruit Extract

The plant extract (KPM018-040) used in this research was obtained from the Natural Product Central Bank at the Korea Research Institute of Bioscience and Biotechnology (Daejeon, Republic of Korea). The plant was collected from Seogwipo-si, Jeju-do, KOREA in 2002. A voucher specimen (KRIB 0000067) is kept in the herbarium of the Korea Research Institute of Bioscience and Biotechnology. The plant (50 g), dried in the shade and powdered, was added to 1 L of Methyl alcohol 99.9% (HPLC grade) and extracted through 30 cycles (40 KHz, 1500 W, 15 min. ultrasonication-120 min. standing per cycle) at room temperature using an ultrasonic extractor (SDN-900H, SD-ULTRASONIC CO., LTD, Seoul, Republic of Korea). After filtration (Qualitative Filter No.100, HYUNDAI MICRO CO., LTD, Seoul, Republic of Korea) and drying under reduced pressure, E. japonica extract (15.44 g) was obtained. For use in laboratory experiments, the obtained extract was dissolved in DMSO at a concentration of 50 mg/mL and further diluted before use.

### 2.2. Cell Culture

3T3-L1 preadipocytes were obtained from ATCC (Manassas, VA, USA) and were maintained in growth media consisting of DMEM-H (Gibco, Paisley, UK) with 25 mM glucose, 3.7 g/L NaHCO3, 1 mM sodium pyruvate, 10% bovine calf serum (*v*/*v*) (BCS; Gibco, Paisley, UK), and 1% penicillin–streptomycin (Gibco, Paisley, UK). The cells were incubated at 37 °C in a humidified atmosphere with 5% CO_2_. When the cells reached 80% confluency in a 100 mm culture dish, they were seeded into a 6-well plate and allowed to grow for an additional 2 days to reach post-confluency before entering the differentiation stage. The 3T3-L1 preadipocytes used in this experiment were at passage numbers ranging from 7 to 13.

### 2.3. Adipogenic Differentiation of 3T3-L1 Preadipocytes

For the experiment, 3T3-L1 preadipocytes were seeded at a density of 1 × 10^5^ cells/well. After 2–3 days, the cells reach 100% confluence. Following this, an additional 2 days are maintained to allow the cells to enter a post-confluence state, where they remain in a cell cycle arrest (G0/G1) state due to contact inhibition and can no longer proliferate. Following the post-confluence state, 3T3-L1 adipocytes were induced to differentiate. The medium was replaced with a medium containing MDI. The MDI used for adipogenic differentiation stands for the cocktail of chemical compounds composed of methylisobutylxanthine (IBMX), dexamethasone, and insulin. The medium consisted of DMEM-H (Gibco, Paisley, UK) with 25 mM glucose, 3.7 g/L NaHCO3, 1 mM sodium pyruvate, 10% fetal bovine serum (*v*/*v*) (FBS; Gibco, Paisley, UK), 1% penicillin-streptomycin (Gibco, Paisley, UK), 1 μg/mL insulin, 0.25 mΜ dexamethasone, 0.5 mM IBMX, and 0.125 nM indomethacin (Sigma, USA, Saint Louis, MO, USA). After the second day of differentiation, the medium was switched to DMEM-H containing 10% FBS and only insulin to promote the maturation of adipocytes. We conducted our experiments using fully differentiated adipocytes on the 8th day of differentiation. When exploring the molecular levels associated with cell cycle arrest during the early stage of differentiation, they were treated with MDI and EJF for 16 and 24 h after 2 days post-confluence. They were then harvested and used in experiments. All media were replaced every 2 days.

### 2.4. Cell Viability Assay

The cytotoxicity of the extract was determined using Cell Counting Kit-8 (CCK-8; Dojindo Molecular Technologies, Kumamoto, Japan). 3T3-L1 cells were seeded into a 96-well plate at 1 × 10^4^ cells/well and maintained for 24 h. Next, the cells were treated with the extract or DMSO for 48 h. Subsequently, CCK solution was added to each well and incubated for 1 h at 37 °C. The absorbance was measured at 450 nm using a microplate reader (Infinite F50; Tecan, Männedorf, Switzerland). For this experiment, EJF was treated at concentrations ranging from 1.58 μg/mL to 200 μg/mL. The group without EJF treatment was treated with 0.004% (*v*/*v*) DMSO.

### 2.5. Oil Red O Staining

According to the experimental design, 3T3-L1 preadipocytes were seeded into 6-well plates and treated with EJF to induce differentiation. On the 6th or 8th day of differentiation, the adipocytes were washed with PBS. Then, they were fixed by treating with 1 mL of 4% paraformaldehyde (Biosesang Inc., Gyeonggi-do, Republic of Korea) for 1 h at room temperature. Subsequently, they were stained with 0.6% Oil Red O solution (Sigma, USA, Saint Louis, MO, USA) at room temperature for 30 min. After staining, the product is washed three times with third-distilled water, and then 1 mL of isopropyl alcohol (Duksan Pure Chemicals, Korea) is added to a 6-well plate. The plate is shaken at 80 rpm for 10 min at room temperature to dissolve the final product (ORO content). Subsequently, the absorbance is measured at 450 nm using a microplate reader in 200 μL aliquots on a 96-well plate.

### 2.6. Real-Time Reverse Transcription Polymerase Chain Reaction (RT-PCR)

3T3-L1 cells were seeded into 6-well plates and treated with EJF according to the experimental design to induce differentiation, after which they were washed once with PBS. Total RNA was isolated using Trizol reagent, and then cDNA libraries were synthesized using the PrimeScript^TM^ RT Reagent Kit (TaKaRa Bio, Kyoto, Japan). For the synthesis of the cDNA library, standard PCR was performed with the following settings: pre-incubation (15 min at 37 °C), annealing (5 min at 50 °C), extension (5 min at 98 °C), and cooling (4 °C). The mRNA expression levels were measured and analyzed using the iCycleriQTM Real-Time PCR Detection System (Bio-Rad Laboratories, Hercules, CA, USA), with detection performed using SYBR Green (TOYOBO, Japan). The specific thermal cycling conditions used for RT-PCR were as follows: pre-incubation (1 min at 95 °C), amplification (15 s at 95 °C, followed by 1 min at 60 °C for 39 cycles), melting (10 s at 95 °C), and cooling (5 s at 72.5 °C). The mRNA expression levels were normalized to those of β-actin and indicated as fold changes compared with a control group. All experiments were performed biologically and technically in triplicate. The primers used for RT-PCR were synthesized by Macrogen (Seoul, Republic of Korea), and their sequences are listed in Table 1.

### 2.7. Western Blot Assay

3T3-L1 cells were lysed in RIPA buffer supplemented with 1× phosphatase cocktail and protease cocktail. The lysate was then mixed with SDS to produce a protein sample, which was boiled at 100 °C for 10 min. Protein quantification was performed using the Bradford method. A 30 μg total protein sample was loaded onto 7.5–15% SDS-polyacrylamide gel for the purpose of separation. Subsequently, the proteins were transferred to nitrocellulose membranes, which were then blocked with a solution of 5% nonfat skim milk in TBS-T buffer (10 mM Tris pH 8.0, 150 mM NaCl, and 0.05% Tween 20) for 1 h at room temperature with shaking at 80 rpm. Primary antibodies were applied and incubated overnight at 4 °C. After three washes with TBS-T buffer, the membranes were incubated with horseradish peroxidase-conjugated secondary antibody for 1 h at room temperature. The protein bands with bound antibodies were detected using an enhanced chemiluminescence kit (GE Healthcare, Buckinghamshire, UK) and visualized using the Fusion Solo Detector. The β-actin and β-tubulin bands were used as a normalization control for the targeted protein bands. The antibodies used for protein detection are listed in Table 2.

### 2.8. Cell Cycle Assay

To evaluate the cell cycle, 3T3-L1 adipocytes treated with MDI and the extract were separated into single cells using Trypsin^TM^ Express Enzyme (TE; Thermofisher Scientific, Waltham, MA, USA), which consists of Trypsin (1X), phenol Red, and EDTA. To examine the cell cycle distribution based on the difference in fluorescence expression levels within a single adipocyte, we detached the adipocytes attached to the well by treating them with TE. Subsequently, we generated single cells through multiple pipetting steps. Single cells were then fixed with cold 95% ethanol and harvested by centrifugation at 3000× *g* for 5 min. The resulting cell pellet was resuspended in a solution containing propidium iodide (100 μg/mL) and Rnase A (100 μg/mL) for staining. Stained cells were incubated at 37 °C for 30 min in a CO_2_ incubator. Cell cycle analysis was performed using an Attune acoustic focusing cytometer (Thermofisher Scientific, Waltham, MA, USA). For the experiment, we set the voltage to 80 for Forward Scatter (FSC) and 260 for Side Scatter (SSC). After gating, we counted 1 × 10^4^ cells and measured them using a 488 nm excitation laser and a 574/26 emission filter (nm).

### 2.9. Statistical Analysis

The experimental results were analyzed using the GraphPad Prism 9.4.1 software (Sandiego, CA, USA) for data processing. All experiments were independently repeated at least three times, and the data are presented as the mean ± standard deviation (SD) of individual measurements within independent experiments. Statistical analysis was conducted using IBM SPSS statistics 25. One-way analysis of variance followed by Tukey’s post hoc comparison test was performed, and statistical significance was set at *p* < 0.05.

## 3. Results

### 3.1. EJF Extract Inhibits Adipogenic Differentiation in 3T3-L1 Preadipocytes

Through this section, we confirmed that EJF inhibits adipogenic differentiation and lipid accumulation in 3T3-L1 adipocytes. Before initiating differentiation, we determined the optimal treatment dose of EJF through cell viability. Since no significant cytotoxicity was observed up to a dose of 50 μg/mL, it was chosen as the highest dose (Figure 1A). Subsequently, adipogenic differentiation was induced over an 8-day period with or without EJF treatment. The scheme used for differentiation is depicted in Figure 1B. As a result, EJF treatment significantly decreased both adipogenic differentiation and lipid accumulation. Furthermore, these effects were found to be dose-dependent. As adipogenic differentiation is regulated by the transcriptional cascade involving factors such as PPARγ and C/EBPα [1], we examined the effect of EJF extract on the expression of adipogenesis-related transcription factors. Adiponectin, secreted by mature adipocytes, can be used as a marker for in vitro differentiation. Therefore, we also investigated the mRNA expression of adiponectin. The protein levels of PPARγ and C/EBPα showed minimal changes at 12.5 and 25 μg/mL EJF concentrations but significantly decreased at 50 μg/mL (Figure 1D,E). Similarly, at the mRNA level, the expression levels of Pparγ, C/ebpα, and Adipoq decreased in a dose-dependent manner (Figure 1F). These findings demonstrate that EJF inhibits the differentiation of preadipocytes into mature adipocytes and impedes lipid accumulation in mature adipocytes.

### 3.2. The Inhibitory Effect of EJF on Adipogenesis Is Most Critical in the Early Stage of Differentiation in 3T3-L1 Preadipocytes

Through this section, we discovered that the inhibitory effect of EJF on adipogenesis is most critical during the early stage of differentiation. EJF inhibits adipogenic differentiation and lipid accumulation, leading to the investigation of the stage at which EJF intervenes. The differentiation of adipocytes can be divided into early stage, middle stage, terminal differentiation depending on the type of cocktail and duration of differentiation. Based on the information below, we divided the differentiation stages into three phases [12]
⬩Early stage: After post-confluence, treatment of MDI initiates differentiation. This stage is characterized by the presence of growth-arrested preadipocytes and mitotic clonal expansion (MCE) phase.⬩Middle stage: After 2 days of MDI treatment, transitioning to media containing only insulin leads to entering the middle stage. In this stage, adipogenic gene expression is initiated, and the accumulation of lipid droplets begins.⬩Late stage: After the 4th of differentiation, adipocytes enter the maturation stage. This stage is characterized by the formation of mature lipid-filled adipocytes.

Firstly, we have designed a scheme to treat EJF at different stages of differentiation (Figure 2A). EJF was administered at a dose of 50 μg/mL. The results revealed a significant difference in the level of differentiation between Group A, which did not undergo EJF treatment, and Group B, which received six days of treatment. Specifically, among the groups treated with EJF for two days (Groups C, E, and G), the differentiation decrease in Group C was approximately 50% lower in terms of ORO content compared to the control group (Group A). However, no significant differentiation decrease was observed in Groups E and G, which received EJF treatment during the middle and late stages, compared to Group C. Furthermore, among the groups treated with EJF for four days (Groups D, F, and H), there was a significant difference in the differentiation between the groups (D and H) that underwent the early stage and the group F that did not undergo the early stage (Figure 2B,C). These results indicate that EJF critically regulates the early stage of differentiation regardless of the treatment duration.

Additionally, the expression level of Pparγ, a master transcription factor that induces adipogenic differentiation, was investigated. When comparing groups with the same total extract treatment duration, similar to the method used for comparing ORO content, the expression of Pparγ was significantly decreased in the groups treated with EJF during the early stage compared to the groups without early-stage treatment (Figure 2D). This is consistent with the trend observed in ORO content. Therefore, our findings suggest that EJF strongly regulates the early stage of adipogenic differentiation, leading to a decrease in lipid accumulation.

### 3.3. EJF Inhibits the Early Stage of Adipogenic Differentiation by Inhibiting MDI-Induced Cell Cycle Progression in 3T3-L1 Preadipocytes

In this section, we discovered that EJF delays cell cycle progression by inducing MDI during the early stage of differentiation. For the progression of differentiation, 3T3-L1 preadipocytes stop their growth through contact inhibition due to inhibitory signals from nearby cells. In preadipocytes with growth cessation, the level of p27 KIP1 protein increases, and the expression levels of cyclin-dependent kinase 4 (CDK4) and cyclin D decrease, resulting in cell cycle arrest in the G0/G1 phase. When MDI is administered in the cell cycle arrest state, MCE (S phase) is initiated after approximately 14 h. Through MCE, the cell cycle of adipocyte is resumed, and adipocytes prepare for lipid formation and accumulation in subsequent stages. Therefore, delaying the initiation of MCE by blocking cell cycle progression may be an efficient method to inhibit adipogenesis [2]. EJF plays a key role in adipogenic differentiation in the early stage, leading to the investigation of its effect on MCE initiation. Based on the MCE initiation point, we have specified certain time points as 16 and 24 h. We conducted the experiment by dividing the groups based on the presence or absence of MDI and EJF as follows: growth-arrested preadipocytes (NC), MDI (+), and EJF-treat groups categorized by concentration (μg/mL). Afterward, we examined the changes in cell cycle distribution and the expression of cell cycle-related genes after EJF treatment, and the results are as follows.
⬩Cell cycle distribution: The NC group induced cell cycle arrest in G0/G1 phase, and the MDI (+) group initiated both cell cycle progression and MCE (S phase). However, after EJF treatment, the percentage of cells in S phase decreased and the percentage of cells in G0/G1 increased (Figure 3A,B).⬩CDK4, Cyclin D: A significant decrease in the expression levels of CDK4 and cyclin D was observed at 16 h in the group treated with EJF compared to that in the group without EJF treatment (Figure 3C–F).⬩p27 KIP1: The MDI (+) group exhibited a decrease in the expression of p27KIP1 compared to the NC group due to the resumption of the cell cycle. However, treatment with EJF significantly increased the expression of p27KIP1, and this observation remained consistent regardless of the time point (Figure 3C,D,G).

This trend was more significant at 16 h than at 24 h. Therefore, this result suggests that EJF treatment delays MDI-induced cell cycle resumption—that is, entry into MCE.

### 3.4. EJF Attenuates IRS1 Stability and Inhibits Its Activation in 3T3-L1 Preadipocytes

In this section, we discover that EJF reduces the stability and activation of IRS1 and subsequently reduces the activation of Akt. When treating adipocytes with MDI for adipogenic differentiation, insulin first binds to the insulin receptor on adipocytes. Subsequently, insulin receptor substrate 1 (IRS1) becomes phosphorylated and activated by the receptor. Activated IRS1 then converts phosphoinositide 3-kinase (PI3K) into phosphatidylinositol 4,5-bisphosphate and phosphatidylinositol 3,4,5-trisphosphate, leading to the phosphorylation of Akt. Akt is then recruited to the cell membrane and becomes activated. Therefore, this signaling cascade directly participates in cellular differentiation and adipogenesis by regulating adipogenesis-related genes. In this process, IRS1, which directly modulates the response to external MDI and subsequent differentiation, exerts a significant influence on the initiation of the early stages of differentiation [11]. EJF plays a key role in adipogenic differentiation in the early stage, leading to the investigation of its effect on IRS1 phosphorylation and activation. We established the hypothesis that EJF treatment would decrease IRS1 phosphorylation and stability, subsequently inhibiting Akt phosphorylation, ultimately leading to the suppression of adipogenesis when inducing differentiation using MDI-containing medium. Phosphorylation of IRS1 was significantly increased in the MDI (+) group compared to the NC group. However, EJF treatment decreased both IRS1 stability and IRS1 phosphorylation in a dose-dependent manner. These results suggest that EJF treatment significantly reduces IRS1 phosphorylation and activation. Next, phosphorylation of Akt was also significantly inhibited in the 50 μg/mL group compared to the MDI (+) group (Figure 4A,B). In summary, EJF inhibits adipogenic differentiation with a strategy to decrease phosphorylation and the stability of IRS1 and inhibit Akt phosphorylation in the early stage of differentiation.

## 4. Discussion

In this study, two mechanisms were proposed to explain the antiobesity effects of EJF during the early stage of adipogenic differentiation. The first mechanism involves the delay in differentiation progression through the inhibition of MCE via cell cycle arrest. The second mechanism involves a decrease in IRS1 stability and reduced phosphorylation of IRS1. Adipogenic differentiation is a highly regulated process involving the regulation of gene expression, induction of transcription factors that drive adipocyte development, and cascades of cell cycle proteins [12]. The MDI-induced differentiation model of murine 3T3-L1 preadipocytes is widely used in obesity research [13]. The early stage of adipogenic differentiation can be largely divided into growth arrest through contact inhibition, MCE, and postmitotic growth arrest stages [14]. When preadipocytes proliferate and reach a high level of confluency, growth contact inhibition occurs within the cells, wherein no more cell growth occurs, and cell cycle arrest continues. In this state, when stimulated by hormones such as insulin and dexamethasone, the cells undergo reentry into the cell cycle, resulting in MCE. After a certain period of time following MCE, another round of growth arrest occurs, resulting in the subsequent attainment of terminal differentiation [15]. During cell cycle arrest and progression, preadipocytes overexpress p27 KIP1 in the first cell cycle arrest and remain in the G1 phase. During the early stages of differentiation, cyclin D is activated and assembled by CDK4. The cyclin D–CDK4 complex acts as a regulator in the early G1 phase of the cell cycle, facilitating the progression of cells from the G1 phase to the S phase [16]. Therefore, inhibiting the progression to MCE by delaying the cell cycle progression of preadipocytes through the regulation of cell cycle-related proteins can be a novel strategy to inhibit adipogenic differentiation. Moreover, when insulin stimulation for the induction of differentiation is provided, insulin binds to the alpha and beta subunits of the insulin receptor in the adipocyte membrane and induces phosphorylation [17], followed by tyrosine phosphorylation of IRS1 [18]. Phosphorylated IRS1 induces the activation of MEK, ERK, and Akt, and these factors regulate adipogenic differentiation [19]. IRS1, which plays such an important role in adipogenic differentiation, is degraded by E3 ligase activated by stimuli such as extract treatment [20]. Hence, reducing the stability or phosphorylation of IRS1 in the early stage of differentiation can be a strategy to suppress adipogenesis.

After being regulated by these upstream factors, adipogenic differentiation is finally regulated by adipogenesis-related transcription factors. PPARγ is a transcription factor that strongly induces adipogenesis by inducing lipid accumulation, morphological changes in adipocytes, adipose tissue-specific gene expression, and cell growth arrest [21]. C/EBPα is one type of CCAAT/enhancer-binding protein and is widely recognized as a transcription factor involved in adipogenic differentiation. Similar to PPARγ, C/EBPα also acts as a master regulator of adipogenic differentiation, maintaining the phenotype of mature adipocytes and cooperatively inducing the expression of adipose tissue-specific genes along with PPARγ [22]. Therefore, we selected PPARγ and C/EBPα as markers for adipogenic differentiation and adiponectin as a marker representing the extent of final adipogenic differentiation. Adiponectin is a hormone secreted by mature adipocytes, and it plays a role in glucose regulation and fatty acid oxidation within the body [23]. However, in the present study, adiponectin was used as a marker to evaluate the degree of adipogenic differentiation and lipid accumulation. EJF treatment regulated the early stage of adipogenic differentiation through these strategies and ultimately reduced the number of mature adipocytes by inhibiting the expression of adipogenesis-related transcription factors throughout the entire process of adipocyte development. In the case of childhood obesity, which is often accompanied by adipocyte hyperplasia, these mechanisms of action of EJF are particularly advantageous for treatment [2].

We suggest further research to supplement our study findings. It is important to identify the bioactive components present in EJF that contribute to the effects observed in this study. Previous studies have reported the presence of euscaphinin and euscapholide in the leaves [24], triterpene acid in the branches [25], and hexacyclic triterpenic acid in the roots [26] of *E. japonica* Kanitz. However, there have been no reports on the components present in the fruit of this plant. Therefore, we intend to analyze and quantify the components present in EJF using MS BPI chromatogram and UPLC-QTOF-MS. Subsequently, we shall conduct in vitro experiments using the fractions obtained from the chromatogram to select one fraction that exhibits similar adipogenic inhibitory activities to those of the original extract. We shall then identify the main component within that fraction using nuclear magnetic resonance [27].

## 5. Conclusions

We discovered that EJF inhibits adipogenesis by inhibiting the early stage of adipogenic differentiation, including the initiation of differentiation. In this regard, we have proposed two major mechanisms for inhibiting adipogenic differentiation induced by MDI. The first mechanism is the delay in entry into MCE due to the induction of cell cycle arrest during the early stage of differentiation. This was demonstrated by cell cycle arrest induced by increased expression of p27KIP1 and decreased expression of CDK4 and Cyclin D compared to the MDI (+) group, resulting in increased G0/G1 distribution. The second is the decreased stability and activation of IRS1 during the initiation stage of differentiation. This was demonstrated by the reduction of IRS1 expression and phosphorylation due to EJF treatment when IRS1 activation was induced by MDI. This resulted in decreased phosphorylation of Akt, leading to the suppression of the expression of PPARγ and C/EBPα, which are master regulators of adipogenic differentiation [28]. As a result, the delay in entry into MCE and decreased stability and activation of IRS1 ultimately reduced the extent of adipogenesis and lipid accumulation during subsequent stages of differentiation. Therefore, through this study, we have demonstrated that EJF effectively inhibits excessive adipocyte formation in the early stage of differentiation, suggesting EJF as a potential candidate for the prevention and treatment of childhood obesity, which is accompanied by adipocyte hyperplasia. Our study can contribute to identifying bioactive components with these effects in EJF to develop a treatment for childhood obesity; thus, it can contribute to treating childhood obesity, which has increased due to the COVID-19 pandemic.

## Figures and Tables

**Figure 1 nutrients-15-03078-f001:**
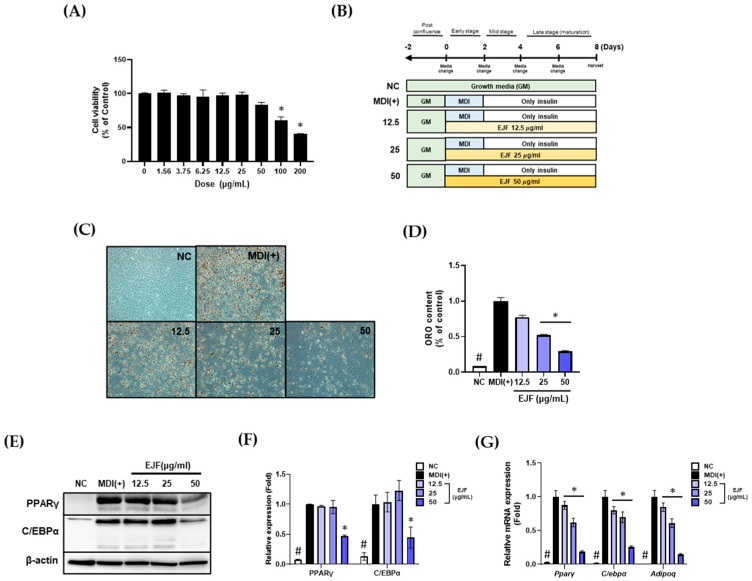
Effects of *Euscaphis japonica* Kanitz fruit (EJF) extract on fully differentiated 3T3-L1 adipocytes. (**A**) A CCK assay was conducted to select a noncytotoxic concentration of EJF before treating 3T3-L1 preadipocytes. Cell viability was considered noncytotoxic if it was >80% of that in the control group, and that concentration was used in the experiments. (**B**) Scheme of adipogenic differentiation for this experiment. (**C**) 3T3-L1 preadipocytes were treated with EJF during the differentiation period. Representative images of each group were obtained through Oil Red O (ORO) staining to evaluate the extent of lipid accumulation. (**D**) Lipid droplets stained by Oil Red O (ORO) staining were quantified by dissolving them in isopropanol and measuring the absorbance at 450 nm. (**E**) Protein expression levels of PPARγ and C/EBPα in fully differentiated 3T3-L1 adipocytes were detected by Western blotting. (**F**) The Western blot bands representing the protein expression levels of PPARγ and C/EBPα were quantified and presented graphically. β-actin values were used to confirm equal loading, and the expression levels of all factors were normalized to those of β-actin. (**G**) mRNA expression levels of Pparγ, C/ebpα, and Adipoq in fully differentiated 3T3-L1 adipocytes were detected using RT-PCR. Equal loading was confirmed by β-actin, and the expression levels of all factors were normalized to β-actin values. All experiments were independently repeated three times, and data are expressed as mean ± SD. * *p* < 0.05 compared with the MDI (+) group. # *p* < 0.01 when compared with the NC group.

**Figure 2 nutrients-15-03078-f002:**
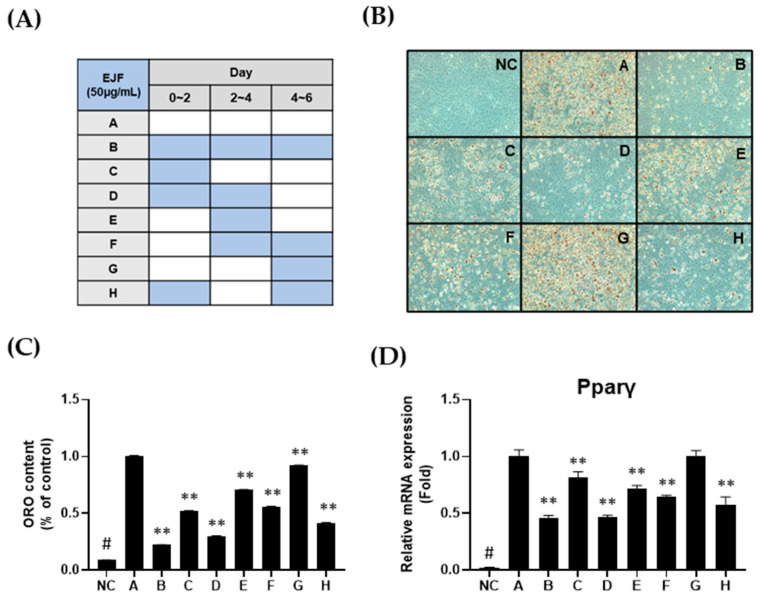
Effects of *Euscaphis japonica* Kanitz fruit (EJF) extract on each stage of adipogenic differentiation. (**A**) Scheme for EJF treatment during the differentiation period of 3T3-L1 preadipocytes. 3T3-L1 preadipocytes were differentiated for 6 days in the differentiation medium containing MDI, depending on the presence or absence of EJF. (**B**) Mature adipocytes differentiated for 6 days were stained for intracellular lipid droplets using Oil Red O (ORO) staining. (**C**) The stained adipocytes were quantified by measuring the absorbance at 450 nm after dissolving them in isopropanol. (**D**) Total RNA was extracted from the 6-day-differentiated adipocytes to measure Pparγ expression using real-time PCR. Equal loading was confirmed using β-actin, and the expression levels of all factors were normalized to those of β-actin. All experiments were performed independently in triplicate, and data are expressed as mean ± SD. ** *p* < 0.01 compared with the A group. # *p* < 0.01 when compared with the NC group.

**Figure 3 nutrients-15-03078-f003:**
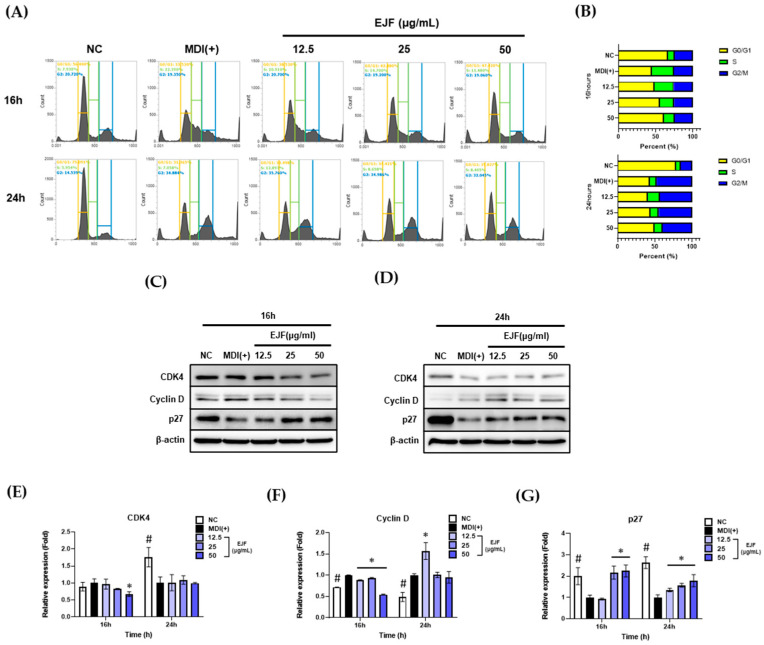
Effects of *Euscaphis japonica* Kanitz fruit (EJF) extract on MDI-induced cell cycle progression in 3T3-L1 preadipocytes. 3T3-L1 preadipocytes were treated with MDI-containing medium for 16 and 24 h, following cell cycle arrest induced by contact inhibition, depending on the presence or absence of EJF. (**A**) 3T3-L1 cells were fixed with 95% ethanol and then treated with propidium iodide. The cell population was determined using a flow cytometer, with a count of 1 × 10^4^ cells per sample. (**B**) The distribution of the quantified cell population was represented graphically. (**C**) After treating the cells with EJF in MDI-containing medium for 16 h, the expression levels of CDK4, cyclin D, and p27 KIP1 proteins were measured by Western blotting. (**D**) After treating the cells with EJF in MDI-containing medium for 24 h, the expression levels of CDK4, cyclin D, and p27 KIP1 proteins were examined by Western blotting. Equal loading of proteins was confirmed by the expression of β-actin. (**E**–**G**) The Western blot bands obtained from cells treated with EJF or control in MDI-containing medium for 16 and 24 h were quantified and presented graphically. The expression levels of all factors were normalized to those of β-actin. All experiments were independently performed in triplicate, and data are expressed as mean ± SD. * *p* < 0.05 compared with the MDI (+) group. # *p* < 0.01 when compared with the NC group.

**Figure 4 nutrients-15-03078-f004:**
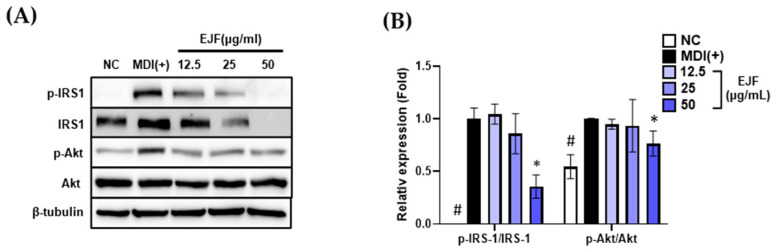
Effects of *Euscaphis japonica* Kanitz fruit (EJF) extract on IRS1 stability and its activation during the early stage of adipogenic differentiation. 3T3-L1 preadipocytes were treated with medium containing MDI for 30 min, depending on the presence or absence of EJF. (**A**) After treating 3T3-L1 cells for 30 min, the expression levels of p-IRS1, IRS1, p-Akt, and Akt proteins were detected by Western blotting. (**B**) The levels of phosphorylated IRS1 and its downstream target Akt phosphorylation were quantified by analyzing Western blot bands and presented graphically. Equal loading was confirmed by β-tubulin, and the expression levels of all factors were normalized to β-tubulin values. All experiments were independently repeated three times, and data were expressed as mean ± SD. * *p* < 0.05 compared with the MDI (+) group. # *p* < 0.01 when compared with the NC group.

**Table 1 nutrients-15-03078-t001:** Primer sequences and corresponding accession numbers used for qPCR.

Gene Name	Accession No.		Sequence
*Adipoq*	NM_009605	Forward	5′-ACCTACGACCAGTATCAGGAAAAG-3′
		Reverse	3′-ACTAAGCTGAAAGTGTGTCGACTG-5′
*C/ebp* *α*	NM_001287523	Forward	5′-TTACAACAGGCCAGGTTTCC-3′
		Reverse	3′-GGCTGGCGACATACAGATCA-5′
*Ppar* *γ*	AB644275	Forward	5′-TTTTCAAGGGTGCCAGTTTC-3′
		Reverse	3′-AATCCTTGGCCCTCTGAGAT-5′
*β-actin*	EF095208	Forward	5′-GACAACGGCTCCGGCATGTGCAAAG-3′
		Reverse	3′-TTCACGGTTGGCCTTAGGGTTCAG-5′

**Table 2 nutrients-15-03078-t002:** Antibody information used for Western blotting.

Gene Name	Company	Product No.	IgG	Conc.
**Primary antibody**				
PPARγ	SCBT	sc-7273	M	1:500
C/EBPα	CST	2295S	R	1:1000
IRS1	CST	2390S	R	1:1000
p-IRS1	Invitrogen	2103503	R	1:1000
Akt	CST	4691S	R	1:1000
p-Akt	CST	4060S	R	1:1000
β-actin	SCBT	sc-47778	M	1:500
β-tubulin	Abcam	ab179513	R	1:1000
**Secondary antibody**				
Mouse	CST	7076S		1:2000
Rabbit	CST	7074S		1:3000

## Data Availability

The data presented in this study are available on request from the corresponding author.

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
