# Peer review of "Euscaphis japonica Kanitz Fruit Exerts Antiobesity Effects by Inhibiting the Early Stage of Adipogenic Differentiation"

_nutrients, 2023, doi:10.3390/nu15143078_

Round 1
Reviewer 1 Report
In this manuscript, authors investigated the effects of Euscaphis japonica Kanitz fruit on adipocyte differentiation and proliferation in vitro, suggesting that EJF could potentially perform anti-obesity effects via inhibition of adipocyte hyperplasia.
1. Please clarify the compassions between groups in all figures, what do those asterisks mean? For fig 2C, why there was no comparison between individual groups?
2. Section 3.2, please add group names in the description of results, it is confusing to match the result part with fig 2. In addition to changes of Pparγ expression, how about C/ebpα and Adipoq?
3. Obesity is associated with low-grade but chronic inflammation. In addition to the inhibitory effects on adipocyte hyperplasia, does EJF have any regulatory effects on adipocyte inflammation?
4. Current experiments conducted in this manuscript only includes normal adipocytes which could be very different from inflamed and obese adipose tissue, it is recommended that more experiments should be conducted in inflamed adipocytes with insulin resistance such as high glucose-treated cells.
5. Please spell out MDI and why chose it as a positive control.
N/A
Author Response
Dear Reviewer,
I am writing to express my sincere gratitude for your valuable comments on my manuscript. Your insightful suggestions have greatly contributed to improving the quality and clarity of my research. I appreciate the time and effort you have dedicated to reviewing our manuscript and providing detailed recommendations for revision. Your guidance and support have been immensely helpful. The corrections have been compiled and uploaded as PDF files.
Please let me know if you have any additional suggestions during the revision process. Your continued guidance will undoubtedly assist me in producing a more comprehensive and robust final manuscript.
Thank you once again.
Best regards,
From the author of 'nutrients-2478028'

Reviewer 2 Report
Dear editor of Nutrients (MDPI) and authors of the manuscript nutrients-2392248, I am pleased to accept this review invitation and take place in the peer review process of this innovative study. Indeed, natural antiobesity agents, primarily from plants, have gained considerable popularity among consumers. This is mainly attributed to the perception that they possess minimal or no side effects compared to pharmaceutical drugs. As a result, there is a significant market for antiobesity agents sourced from natural ingredients, which have been extensively investigated for their effectiveness and safety. Considering these factors, choosing to focus on the fruit of Euscaphis japonica Kanitz as a potential candidate for a natural antiobesity agent is brilliant. Although previous studies have reported the antiobesity effects of the leaf of E. japonica Kanitz, the underlying molecular mechanisms responsible for adipogenic differentiation have not been fully elucidated. Furthermore, the antiobesity effects of the fruit itself have not yet been documented. I did not find similar publications in PubMed. Before considering this text for publication, Nutrients must address certain limitations.
Please, dear authors, resolve all the commentaries of the annexed PDF document. These commentaries include the text structure and methodology suggestions.
Despite being asked for significant revisions, I wholeheartedly recommend the publication of this manuscript in the journal Nutrients. I sincerely thank the esteemed authors for their invaluable effort in reviewing and enhancing this manuscript, which encompasses a wealth of high-quality scientific information.
With best regards,
The Reviewer.

The level of English proficiency needs enhancement, although I believe that MDPI's English editing service, implemented during the final stages of the revision process, will effectively address all the required corrections.
Author Response

(The authors gave the same response as above.)

Round 2
Reviewer 1 Report
Authors have addressed all my concerns.
N/A
Reviewer 2 Report
Esteemed authors, I extend my heartfelt gratitude for your invaluable dedication in considering my suggestions and enhancing the work. Your willingness to invest time and effort in incorporating my input is genuinely appreciated.
I wholeheartedly endorse the publication of your manuscript.
While the manuscript requires some moderate English corrections, I believe that MDPI's final service English team will be able to address all the concerns adequately.